# Effect of closed and permanent stoma on disease course, psychological well-being and working capacity in Swiss IBD cohort study patients

Rahel Bianchi[1]*, Barry Mamadou-Pathé[2], Roland von Känel[3], René Roth[1,4], Philipp Schreiner[1], Jean-Benoit Rossel[2], Sabine Burk[1], Babara Dora[1], Patrizia Kloth[1], Andreas Rickenbacher[1], Matthias Turina[1], Thomas Greuter[1,5], Benjamin Misselwitz[6], Michael Scharl[1], Gerhard Rogler[1], Luc Biedermann[1], on behalf of the or the Swiss IBD cohort study[¶]

1 Department of Gastroenterology and Hepatology, University Hospital Zurich, University of Zurich, Zurich, Switzerland, 2 Department of Epidemiology and Health System, Unisanté, Lausanne, Switzerland, 3 Department of Consultation-Liaison Psychiatry and Psychosomatic Medicine, University Hospital Zurich, Zurich, Switzerland, 4 Department of Internal Medicine, Limmattalspital, Schlieren, Switzerland, 5 Department of Gastroenterology and Hepatology, CHUV, Lausanne, Switzerland, 6 Department of Visceral Medicine, University Hospital Bern, Bern, Switzerland

¶ Membership of the Swiss IBD cohort study is provided in the Acknowledgements.
* rahel.bianchi@bluewin.ch

**Data Availability Statement:** All relevant data are within the paper and its Supporting Information files.

## Abstract

### Background

Little is known about the impact of ostomy formation in inflammatory bowel disease patients on course of disease, psychological well-being, quality of life and working capacity.

### Methods

We analyzed patients over a follow-up of up to 16 years in the Swiss inflammatory bowel disease cohort study (SIBDCS) with prospective data collection. We compared Ulcerative colitis and Crohn's disease patients with and without ostomy as well as permanent and closed stoma formation before and after surgery, investigating disease activity, psychological well-being and working capacity in a case-control design.

### Results

Of 3825 SIBDCS patients, 176 with ostomy were included in the study and matched with 176 patients without ostomy using propensity score, equaling 352 patients for the analysis. As expected, we observed a lower mean and maximal disease activity in patients after stoma surgery compared with control patients without stoma. Overall, psychological wellbeing in patients with stomas vs. controls as well as patients with permanent vs. closed stoma was similar in terms of disease-specific quality of life (total score of the Inflammatory Bowel Disease Quality of Life questionnaire), psychological distress (total score of the Hospital Anxiety and Depression Scale), and stress at work (effort-reward-imbalance ratio), with the

**Funding:** G.R., Grant No.310030-120312, Swiss National Science Foundation, https://www.snf.ch/de The SIBDCS in general, BASEC-Number 2018-02068, Swiss National Science Foundation, https://www.snf.ch/de The funders had no role in study design, data collection and analysis, decision to publish, or preparation of the manuscript.

**Competing interests:** The authors have declared that no competing interests exist.

exception of a higher Posttraumatic Diagnostic Scale total score in patient with vs. without stoma. Compared to IBD patients without stoma, the adverse impact on working capacity in overall stoma IBD patients appeared to be modest. However we observe a significantly higher reduction in working capacity in permanent vs. closed stoma in CD but not UC patients.

## Conclusion

As to be expected, IBD patients may benefit from closed and permanent stoma application. Stoma surgery appears to only modestly impact working capacity. Importantly, stoma surgery was not associated with adverse psychological outcomes, with comparable psychological well-being regardless of presence and type of stoma.

## Introduction

The etiology of Inflammatory Bowel Disease (IBD) consisting of Crohn's Disease (CD) and Ulcerative Colitis (UC) [1] is incompletely understood and the global burden is high. It is estimated that approximately 3.1 Million Americans [2] and 2.5 Million Europeans suffer from IBD [3]. Numerous pathogenetic factors are involved in these complex chronic immune-mediated diseases, including genetic predisposition, epithelial barrier effects, environmental factors and a dysregulated immune response [4]. Overall around 50% of IBD patients will need surgery during their lifetime [5]. Mostly it is necessary for difficult-to-treat IBD patients refractory to a variety of biological (i.e. all monoclonal antibodies; biologics) and non-biological (i.e. immunomodulators, corticosteroids, mesalamine, calcineurin-inhibitors, small molecules) agents, or due to complications, such as toxic megacolon, perforation, hemorrhage or peritonitis [6]. Surgical treatment encompass a wide variety of options from strictureplasty, segmental colonic, small bowel or ileo-cecal resection to complete proctocolectomy.

Complete proctocolectomy representing the standard procedure for refractory UC, nowadays in the vast majority of cases performed in a three-step (less frequently two-step) procedure, therefore encompassing closed stoma for a period of several months [7,8] with multiple anastomosis possibilities, latter being the current method of choice [9]. In UC, the risk of colectomy was shown to be 4.1%, 6.4% and 14.4% after 5, 10 and 20 years respectively with a decreasing trend according to a recent investigation in Switzerland [10] with comparable results in North-American studies [11,12]. In refractory UC the most frequent surgical procedures are ileal pouch anastomosis (IPAA) and proctocolectomy with ileostomy [13]. However, also in CD and despite advances in anastomosis techniques stoma rates are still considerable with around 1.51–1.9 stomas per 100 person years with permanent stoma formation at a stable level in recent years [14,15].

Studies have shown conflicting data on quality of life in patients after ostomy [16–19]. Moreover, stoma patients with CD have been found to display high rates of anxiety and depression without receiving sufficient psychological support [20] and stoma formation was identified as a risk factor for the development of these problems in IBD patients in general [21]. Moreover not only ostomy formation has been identified as a risk factor for post-traumatic stress symptoms but also surgery, hospitalizations, and disease severity in general as well with medical procedures, surgery being two of the main five factors identified by an American, qualitative study [22]. Additionally, studies have indicated CD-induced posttraumatic stress worsening the disease course [23] and a negative effect on work ability of IBD patients [24,25]; yet further research is needed to determine the impact of stoma surgery.

Therefore, we aimed to investigate in a case control study design whether there are differences in the course of the disease, clinical symptoms, medication use, psychological well-being, quality of life and work ability in IBD patients with vs. without ostomy as well as pre vs. post relocation surgery in those stoma patients with later closed ostomy formation.

## Methods and material

### Patient data and study design

All patients participating in the Swiss IBD Cohort Study (SIBDCS) with history of stoma surgery were eligible to be analyzed in our study. More details on the goals, structure and methodology of this multicenter prospective observational population-based cohort study also with regards to the validated outcome scores included in the inclusion/annual follow-up questionnaires for physicians and patients have been described elsewhere [26].

We conducted a multi-center, case-control study including patients diagnosed with either CD or UC that participated in the Swiss IBD cohort study with prospective and standardized data collection since 2005. Three groups of patients were defined and compared using Wilcoxon- and Chi-Square-Test. The first group consisted of patients who received a permanent stoma, the second group included patients whose stoma has been closed and the third group comprised of matched patients according to diagnosis, age, gender, disease duration and disease severitiy, extraintestinal manifestations (EIM) such as peripheral arthritis/artralgia, uveitis/iritis, pyoderma gangrenosum, erythema nodosum, aphthous oral ulcers, ankylosing spondylitis, sacroiliitis, and primary sclerosing cholangitis as well as a composite of typical complications associated with IBD (including gallstone formation, anemia (not due to drug adverse events), deep venous thrombosis, colorectal cancer, colonic dysplasia, pulmonary embolism, intestinal lymphoma, malabsorption syndrome, massive hemorrhage, growth failure, osteopenia/osteoporosis, perforation/peritonitis and nephrolithiasis), therapies and family history but no stoma surgery in a 1:1 ratio as a control group. Only patients undergoing construction (and removal in closed stoma group) during the prospective follow-up time in the SIBDCS were considered in our analysis. Patients with stoma surgery prior to inclusion in the SIBDCS were excluded. Patients were eligible if they had completed a minimum of one annually follow-up questionnaire for medical, psychological and work related data before, one whilst stoma *in situ* as well as after bowel reconnecting surgery in patients with surgical removal of stoma. Patients undergoing Ileal Pouch Anal Anastomosis (IPAA) prior to the first follow-up visit were also included. For the comparison between the control group and the stoma group, stoma had to be in place at the time of the comparison and at least three follow-ups had to be completed at year 1, 3 and 5. Disease severity was quantified using Crohn's Disease Activity Index (CDAI) [27] for CD and Modified Truelove and Witts Activity Index (MTWAI) [28] for UC. CDAI and MTWAI were measured when the patients were included in the study as well as during annual follow up. The term severe disease was defined for CD and UC patients reaching the individual maximum value of the respective index. In the standardized SIBDCS follow-up no assessment of a modified CDAI specifically for CD patients with a stoma is designated, such as for instance a the modified score proposed by Ishida et al. which does not include liquid or soft stools often difficult to assess in patients after ostomy [29]. However, as CDAI is the most frequently used score in clinical trials [30] and has been used to assess disease reoccurrence also in other instances with surgically altered stool frequency including after ileocolic resection [31,32], we considered it to be the most feasible tool to measure disease activity after ostomy. The IBD Quality of Life questionnaire (IBDQ) [33], the Hospital Anxiety and Depression Scale (HADS) [34], the Posttraumatic Diagnostic Scale (PDS) [35] and the effort-reward-imbalance ratio [36] were used to asses psychological

wellbeing. The IBDQ is a standardized questionnaire assessing quality of life specifically in IBD patients based on their symptoms with a higher score demonstrating a lower disease related quality of life or an insufficient disease control. The HADS is a validated psychometric instrument with 14 items to measure anxiety and depression with a higher total score indicating a greater level of psychological distress in the previous 7 days. The PDS measures the intensity of 17 symptoms of posttraumatic stress in the previous months according to the Diagnostic and Statistical Manual of Mental Disorders, fourth edition (DSM-IV) [37],as previously outlined [23]. Individual items of the PDS were anchored to the experience of IBD as the defining traumatic event. The PDS avoidance score assesses the need to avoid stimuli associated with the traumatic event. The PDS re-experiencing score measures for example nightmares or the feeling the trauma is happening again. A higher score indicates more recollections of the traumatic event. The PDS hyperarousal score measures for instance startle reactions. The effort-reward-imbalance ratio is an index of effort spent and the resulting reward obtained at work based on Siegrist's Effort-Reward Imbalance model [38]. A higher effort-reward-imbalance ratio has been associated with an increased risk of various health and psychological problems [39].

## Statistical analysis

Chi-square-Test [40] and Wilcoxon-Test [41] were used to compare patients with closed stoma to patients with permanent stoma. The same analyses were used for the comparison between stoma patients and patients without ostomy. In order to remove bias due to confounding variables in this observational study we used propensity score matching to select all subjects in the control group. The following information was included to match the control group and the ostomy group: diagnosis, age, gender, disease duration, occurrence of complication, prior intestinal surgery or EIM, therapy with 5-ASA, antibiotics, immuno-modulators, biologics or steroids, nonsteroidal anti-inflammatory drugs (NSAID) intake, disease activity index, family history of IBD, occurrence of "other medical history" (coeliac disease, Tuberculosis (TBC), organ transplantation or psoriasis). A p-value below 0.05 was considered significant.

## Ethical approval

The SIBDCS has been approved by the Ethics Committee (BASEC Number 2018–02068). Patients gave a written informed consent before participating in the cohort study agreeing to data collection and research analysis. Moreover, the current study has been reviewed and accepted by the scientific board of SIBDCS.

## Results

### Patient characteristics

Amongst 3825 SIBDCS patients, 176 matched the inclusion criteria including stoma creation surgery during prospective SIBDCS follow-up and thus could be analyzed and matched to 176 SIBDCS patients without ostomy formation (54% and 46% with CD and UC, respectively), equaling a total of 352 patients analyzed. In 111 patients ostomy was permanent whereas in 65 a subsequent surgery for ostomy closure was performed. CD patients were diagnosed later after first IBD-associated symptoms than UC patients with a median disease duration of 14.8 years in CD and 8.4 years in UC respectively. The fraction of patients experiencing EIM was evenly distributed. Regarding medical therapy more immune modulators and biologics were used in CD patients with stoma. In the stoma group no significant differences could be seen in the variables at baseline when comparing permanent to closed stoma. (Table 1).

**Table 1. Patient characteristics in different groups.**

| | Permanent stoma | Closed stoma | p-value (chi2) | CD | UC | p-value (chi2) | Stoma | No stoma | p-value (chi2) |
|---|---|---|---|---|---|---|---|---|---|
| **Number of patients** | **111** | **65** | | **93** | **83** | | **176** | **176** | |
| **Diagnosis at ostomy time** | | | | | | | | | |
| Crohn | 62 (55.9%) | 31 (47.7%) | 0.295 | | | | 94 (53.4%) | 95 (54%) | 0.915 |
| UC/IBDU | 49 (44.1%) | 34(52.3%) | | | | | 82 (46.6%) | 81 (46%) | |
| **Gender** | | | | | | | | | |
| Male | 64 (57.7%) | 34(52.3%) | 0.490 | 49 (52.7%) | 49 (59.04%) | 0.397 | 98 (55.7%) | 104(59.1%) | 0.518 |
| Female | 47 (42.3%) | 31(47.7%) | | 44 (47.3%) | 34 (41%) | | 78 (44.3%) | 72 (40.9%) | |
| **Family history of IBD (y/n)** | 18 (16.2%) | 4 (6.1%) | 0.051 | 11 (11.8%) | 11 (13.25%) | 0.775 | 22 (12.5%) | 22 (12.5%) | 1.00 |
| **NSAID intake (y/n)** | 16 (14.4%) | 8 (12.3%) | 0.694 | 14 (15.05%) | 10 (12.05%) | 0.562 | 24 (13.6%) | 30 (17%) | 0.375 |
| **Other medical history (y/n)** | 31 (28%) | 18 (27.7%) | 0.973 | 28 (30.1%) | 21 (25.3%) | 0.478 | 49 (27.8%) | 52 (29.5%) | 0.724 |
| **Age at ostomy time** Median, q25 –q75, min–max | 43, 31–57 6–81 | 40, 29–51 14–76 | 0.097 | 42, 30–52 6–81 | 41, 30–53 14–79 | **0.0002** | 41.6, 30.6 – 52.7 6–81 | 42.6, 31.7– 55.4 9–88 | 0.6645 |
| **Disease duration at ostomy time** Median, q25 –q75, min–max | 11.6, 6–18 1–44 | 11, 6–17 1–33 | 0.563 | 14, 8–23 1–44 | 8, 4–14 1–43 | 0.1045 | 11.5, 6–18 1–44 | 12.4, 4–2 0–53 | 0.8385 |
| **Occurrence of** | | | | | | | | | |
| Complications | 92 (82.9%) | 56(86.1%) | 0.567 | 76 (81.7%) | 72 (86.7%) | 0.363 | 148 (84.1%) | 132 (75%) | **0.034** |
| Extra-intestinal manifestation | 70 (63.1%) | 47 (72.3%) | 0.210 | 62 (66.7%) | 55 (62.3%) | 0.955 | 122(69.3%) | | 0.568 |
| Hospitalization related to IBD | 67 (60.4%) | 42(64.6%) | 0.575 | 59 (63.4%) | 50 (60.2%) | 0.663 | 117 (66.5%) 109 (62%) | 58 (33%) | **0.001** |
| **Therapy with…** | | | | | | | | | |
| 5-ASA | 88 (79.3%) | 51(78.5%) | 0.898 | 60 (64.5%) | 79 (95.2%) | **0.001** | 139 (79%) | 159(90.3%) | **0.003** |
| Antibiotics | 87 (78.4%) | 46(70.8%) | 0.257 | 72 77.4% | 61(73.5%) | 0.545 | 133 (75.6%) | 114(64.8%) | **0.027** |
| Immuno-modula- tors | 101 (91%) | 59(90.8%) | 0.961 | 89 (95.7%) | 71(85.5%) | **0.02** | 160 (91%) | 174(98.9%) | **0.001** |
| Biologics | 81 (73%) | 54(83.1%) | 0.126 | 77 (82.8%) | 58(69.9%) | **0.043** | 135 (76.7%) | 142(80.7%) | 0.362 |
| Steroids | 104 (93.7%) | 63(96.9%) | 0.348 | 89(95.7%) | 78 (94%) | 0.604 | 167 (94.9%) | 172(97.7%) | 0.158 |

Comparison of the baseline variables in stoma and non stoma patients, patients with permanent or closed stoma as well as patients with CD and UC.

## Course of disease

First we aimed to investigate disease activity after ostomy in the control group and both ostomy-groups (i.e. the permanent and closed), as well as according to subtype of IBD. The mean CDAI after stoma construction surgery was significantly lower compared to CD patients in the control group not receiving ostomy. Similarly, the maximal CDAI was significantly reduced after ostomy. In contrast, we did not observe such benefits in MTWAI in UC patients with stoma in comparison to the control group (Fig 1).

Furthermore, we identified a lower fraction of EIMs in IBD patients undergoing ostomy in comparison to patients without stoma despite matching the control groups according to several baseline characteristics including previous EIM. Regarding medical therapy, a lower fraction of treatment with 5-ASA, immuno-modulators, biologics as well as steroids was observed in IBD patients with stoma. Moreover, we did not find stoma application to be associated with more complications, however an increase in hospitalization rates was observed. (Table 1).

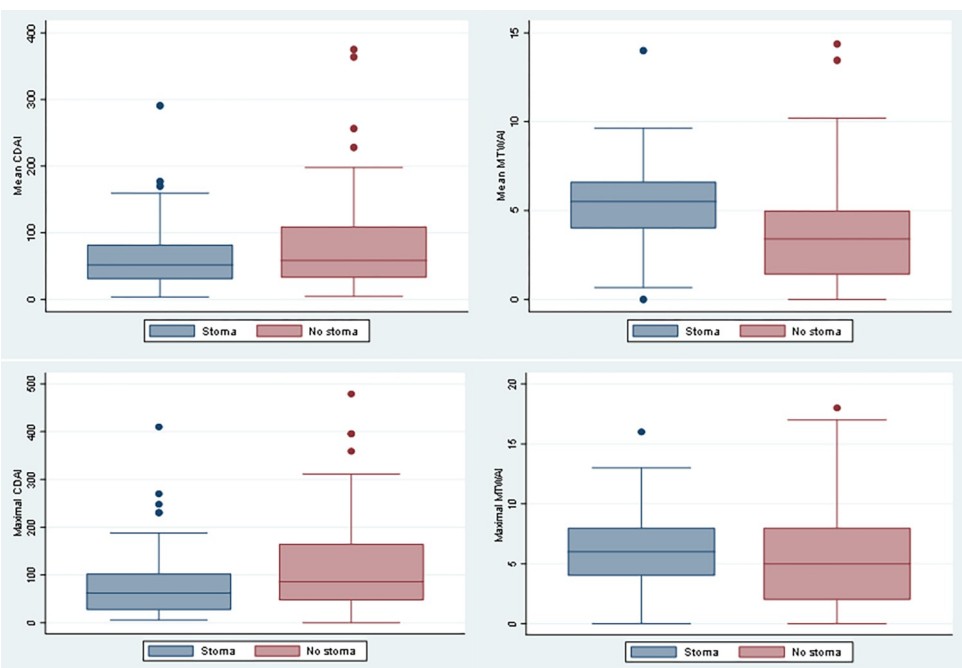

**Fig 1. Comparison of mean and maximal Crohn's Disease Activity Index (CDAI) and Modified Truelove and Witts Activity Index (MTWAI) in patients with and without stoma.**

Next, we aimed to investigate the fraction of patients with permanent or closed stoma and whether stoma type associates with the course of disease and complications. We therefore compared 111 patients with permanent stoma to 65 patients whose stoma has been closed. Of the 111 patients with permanent stoma 62 (55.9%) were experiencing CD and 49 (44.1%) from UC. In the closed stoma group 31 (47.7%) CD and 34 (52.3%) UC patients, respectively, were included. As expected after colectomy in UC, we found a significantly lower median UC disease activity in patients with permanent stoma. Similar results were observed regarding maximal MTWAI after ostomy (Fig 2). Furthermore, we found, that UC patients with closed stoma were significantly more often treated with antibiotics than participants with permanent stoma (2% vs. 17.7%, p = 0.012, S1 Fig). In contrast in CD, no significant decrease in mean nor maximal disease activity was observed (Fig 2).

Regarding medication use, hospitalization, EIM as well as complications permanent stoma application did not appear to be associated with inferior outcomes compared to closed ostomy but stoma patients in general where more often hospitalized than patients without (Table 1).

## Psychological wellbeing

To assess the long-term impact on closed/permanent ostomy on psychological well-being, we aimed to compare the psychological wellbeing indices before and after ostomy and investigate whether there are differences in the evolution of these indices in patients with permanent vs. closed ostomy. When comparing patients with stoma to patients without stoma no differences could be found in the total scores of IBDQ, HADS and effort-reward ratio. However, patients with stoma scored higher in PDS total score. In the patient population with stoma, we did not observe significant differences in the total scores of IBDQ, HADS, PDS and effort-reward-imbalance ratio between patients with persistent vs. closed stoma either. Of note, we observed a lower PDS avoidance score in patients with permanent stoma after ostomy with an increase

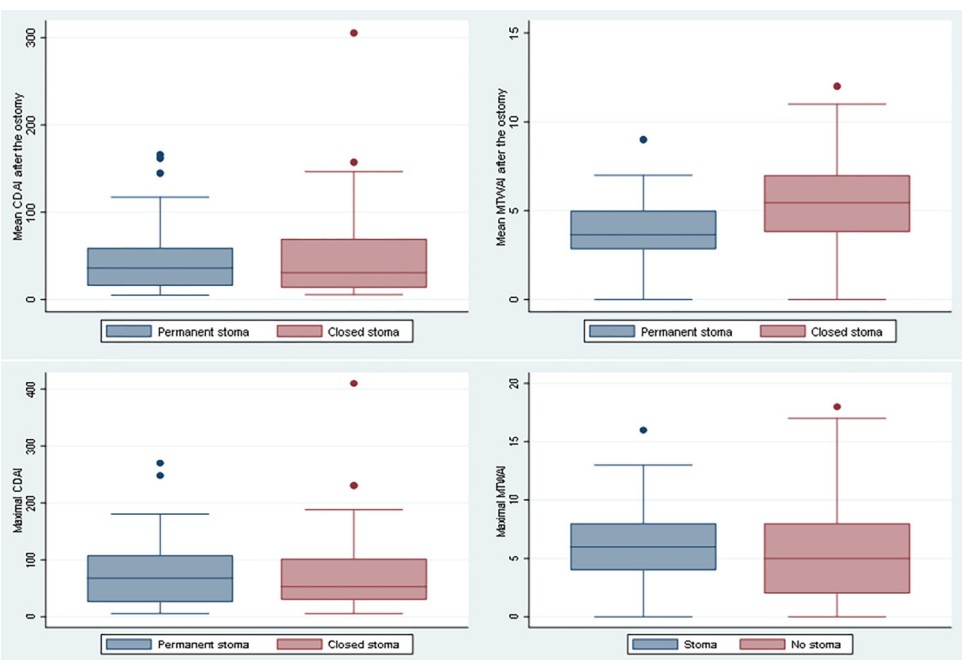

**Fig 2. Mean and maximal disease activity in permanent vs. closed stoma in patients with Crohn's Disease and patients with Ulcerative colitis.**

in contrast in the closed stoma group (before 5 vs. 3, p = 0.036, after 3.8 vs. 4.5, p = 0.672, S2 Fig). Looking at CD and UC patients separately, again no significant differences were observed in the total scores in line with the above-mentioned data (Fig 3A–3D). However a decrease in PDS avoidance score could be observed in CD patients with permanent stoma, after ostomy with the opposite in the closed ostomy group before (5 vs. 1, p = 0.018, after 3.8 vs. 4, p = 0.83, S3 Fig) Regarding the PDS Total score before vs. after ostomy in CD patients with closed ostomy, we found a more than three-fold increase whereas a decrease was observed in those with permanent ostomy (PDS Total Score in patients with closed vs. permanent ostomy, respectively, before 12.2 vs. 4, p = 0.049, and after 8.1 vs. 14, p = 0.99 ostomy, S4 Fig). Moreover, regarding the IBDQ social function score, a significant difference in CD patients with permanent or closed ostomy alone was observed before but not after ostomy (before 23.3 vs. 34, p = 0.001, after 25.5 vs. 33.7, p = 0.06, S5 Fig). Nevertheless, when comparing the overall IBD population with stoma to the patient group without stoma IBDQ social function scores were lower in the ostomy group (27.8 vs. 30.7, p = 0.035 and 30 vs. 34, p = 0.004, S6 Fig). Furthermore, we also observed higher posttraumatic stress (Fig 3C) with increased symptoms associated with ostomy (4 vs. 2.4, p = 0.005, S7 Fig) in patients undergoing ostomy in comparison to those without stoma. In line with that, a significantly higher re-experiencing score was found in comparison to patients without ostomy (2.3 vs. 1.3, p = 0.018, S8 Fig).

In summary it can be stated that IBDQ, HADS and effort reward ratio do not differ between either the stoma group an the control group, the persistent and closed ostomy group nor UC and CD patients. However, PTSD symptoms are higher in stoma patients in general and especially in the permanent stoma group. Similarly, IBDQ social function score decreases after ostomy. Moreover, especially CD patients show noticeable changed in scores after ostomy. This includes lower PDS avoidance score and PDS total score in patients with permanent stoma.

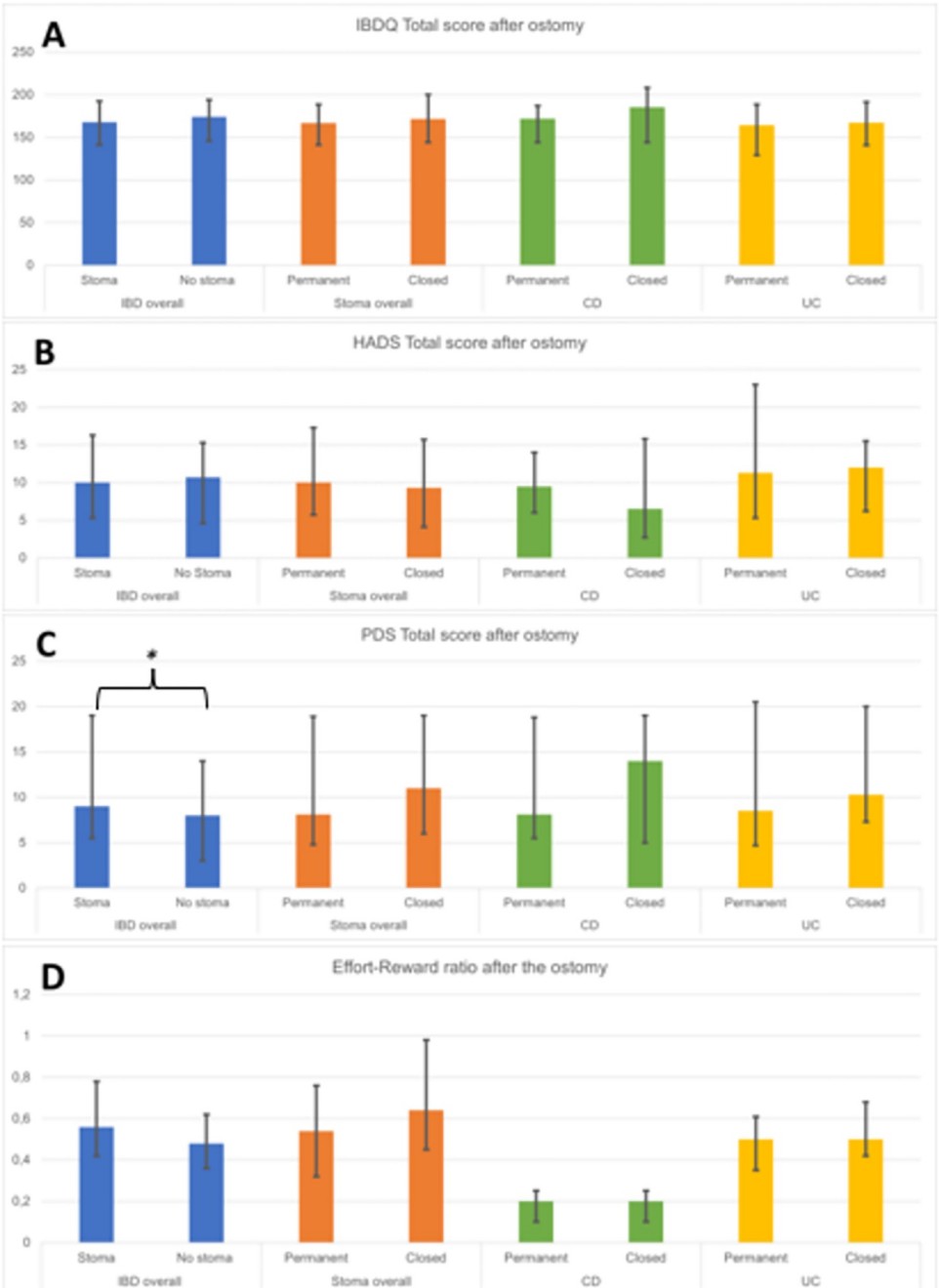

**Fig 3. Comparison of different psychological scores to assess wellbeing of IBD patients.** A) Comparison of median and minimal and maximal scores of Inflammatory Bowel Disease Quality of Life Questionnaire Total Score (IBDQ Total Score) in different patient groups as an indicator of health related quality of life. B) Comparison of median and minimal and maximal scores of Hospital Anxiety and Depression Scale (HADS) Total Score in different patient groups as an indicator of anxiety. C) Comparison of median and minimal and maximal scores of Posttraumatic Diagnostic Scale (PDS) Total Score in different patient groups as an indicator of Post-Traumatic Stress Disorder. D) Comparison of median and minimal and maximal scores of Effort-Reward Ratio in different patient groups.

## Working capacity and invalidity pension

Comparing working capacity overall, we found, that 63.8% of patients with stoma were working at any point during the SIBDCS compared to 71.3% of the IBD patients without stoma

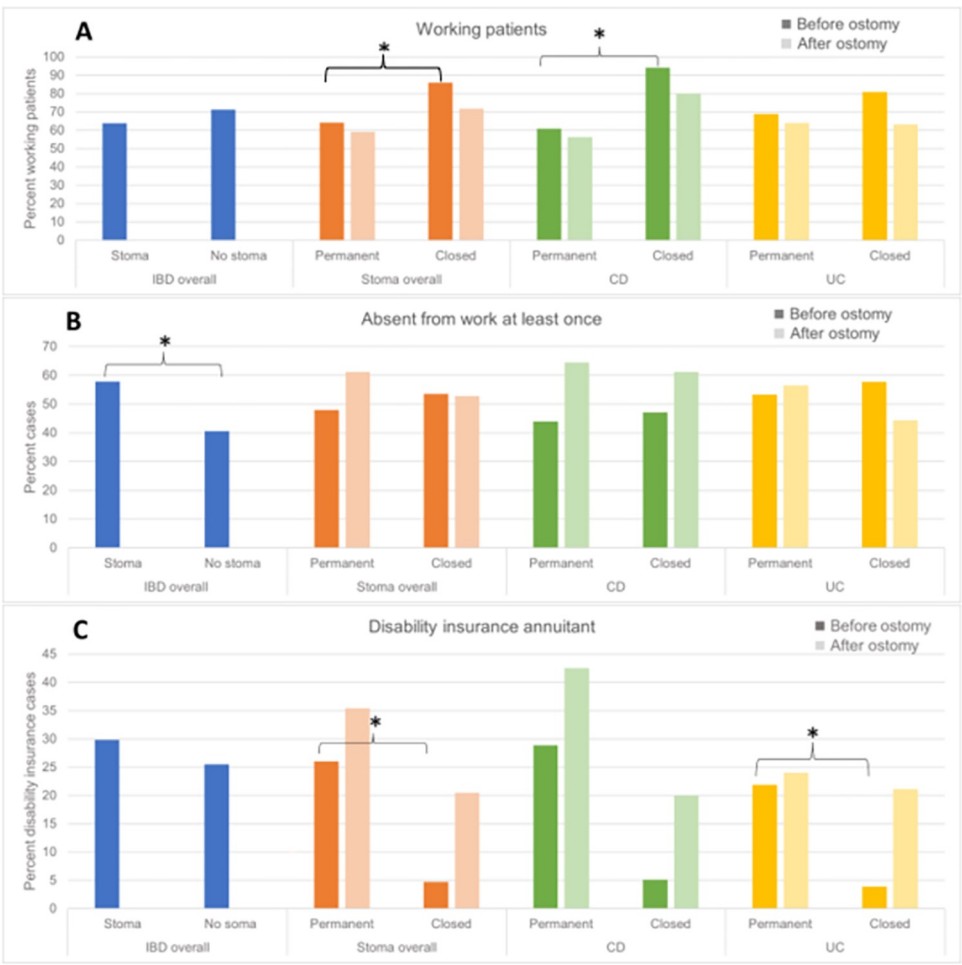

**Fig 4. Working capacity and invalidity pension related data in different groups.** A) Working patients in different patient groups. B) Number of patients in different groups absent from work at least once during participation in the Swiss Inflammatory Bowel Disease Cohort Study. C) Patients in different groups with disability benefits.

(Fig 4A). Any absenteeism from work at least once during the follow-up in the cohort study was more frequently reported in IBD patients with vs. without stoma (Fig 4B).

A lower fraction of working patients prior to stoma construction surgery was identified in the permanent stoma group, compared to the closed stoma group (Fig 4A). Congruently, a higher fraction of patients receiving disability benefits was found in patients with permanent vs. closed stoma prior to the time of stoma construction surgery (Fig 4C). After ostomy an overall lower fraction of working patients could be observed in permanent as well as closed stoma patients with an even larger decrease of working patients in the closed stoma group (Fig 4A). Looking at CD and UC in separate, an analogous trend was observed.

Furthermore, the fraction of patients receiving a disability insurance was higher amongst patients with permanent stoma and increased after stoma surgery. In contrast, this fraction was even lower in patients with closed stoma compared to the matched IBD patients without stoma surgery (Fig 4C).

In addition, more patients in the permanent stoma groups reported any absence from work at the follow-up visits, specifically in CD (Fig 4B).

## Discussion

In our study we aimed to investigate the impact on closed and permanent stoma surgery in patients with IBD on relevant patient-centered outcome parameters, including clinical disease characteristics, psychological parameters and working capacity. Our results indicate, that stoma formation in general–regardless of closed vs. permanent ostomy and against what several patients and physicians alike might expect–is not a priori associated with an adverse psychological outcome. Also, in appropriate CD patients diagnosed with severe disease, these patients may benefit from stoma application to reduce disease activity. However, this study design evidently was unable to provide a definite guidance on whether permanent or closed stoma represents the favored therapeutic strategy in these type of patients. On the one hand, we could did not distinguish between (or respectively perform separate analysis according to) the primary driving reason(s) for stoma closure. This primary motive often is not fully declared in medical records, and not seldomly, there may be a mixed bag of factors, driving the treatings physicians and affected patients towards the decision of preserving vs. reverting ostomy. In CD patients a high recurrence rate most commonly at the side of the anastomosis and/or ostomy is known [42]. It could therefore be speculated that specifically patients without inflammatory recurrence of CD could benefit more from closed ostomy, while these patients at the same time are more likely to have more favorable overall outcome parameters than their counterparts with recurrence.

Moreover, we have observed a significant decrease in medication and EIM in IBD patients in general after stoma application indicating a potential reduction in the severity of disease course. Our results are in line with those from Goudet et al. [43], who previously reported a reduction of EIM in UC after proctocolectomy, and also those from our own group [44]. It can therefore be concluded that patients with CD potentially could benefit from ostomy formation especially in those patients, with complicating course of disease, persistent activity of disease or EIM not being under adequate control and those patients with treatment-related adverse events.

This would be in line with the British Society of Gastroenterology consensus guidelines [6] as well as the Ulcerative Colitis Practice Guidelines in Adults [45] stating that surgery represents a valuable treatment option in case of failing medical therapy or intolerable medical side effects. Furthermore and as to be expected in a disease limited to the colon, we confirmed that also patients with UC can benefit from permanent stoma application as disease activity could be reduced significantly.

Of crucial note in this regard is the fact that permanent ostomy was non inferior to closed ostomy in IBD patients overall regarding medication use, hospitalization, EIM as well as complications.

As a limitation of our design enabling a matching process including amongst others disease severity and prior complications, we cannot exclude, that patients in both stoma groups a priori represent a fraction of a more difficult-to-treat patient population, acknowledging that any matching per se is likely to remain imperfect. Therefore, ever being in need of stoma formation surgery (be it closed or permanent) might constitute a surrogate parameter for a more severe course of disease, higher likelihood of complications and disability. Having this in mind and considering the relatively favorable results observed in our outcome parameters of interest in stoma vs. non-stoma IBD patients, one might assume, that the overall potential benefit of closed and permanent stoma formation may even have been underestimated.

As of today, to the best of our knowledge no study has been conducted comparing permanent with closed stoma in this particular patient group. It is generally well established that IBD may have an adverse impact on quality of life, however data after ostomy is controversial [16–

19]. According to our data permanent stoma in both UC and CD compared to no stoma surgery at all as well as permanent vs. closed stoma surgery does not to translate into inferior or unfavorable outcomes with regard to clinical parameters, psychological well-being and disability. This indicates that IBD patients with an underlying robust indication for stoma surgery, either persistence of stoma or stoma closure surgery, do not appear to have a sustainably impact future psychological wellbeing. This contradicts previous studies indicating surgery to be a trigger for PTSD [22] and showing that patients with higher PTSD symptoms have more likely had surgery [46]. This correlates with the finding that patients in the SIBDCS with ostomy scored higher in the PDS total score, showed a higher avoidance of stimuli associated with ostomy and a higher re-experiencing score in the SIBDCS. This could however also be due to a relatively more severe course of disease (stoma as a surrogate for a more debilitating disease course, imperfect matching; as mentioned above) which has also been shown to impact severity of PTSD symptoms [46]. It has also to be stressed that none of the studies focused specifically on ostomy and that we were able to show a decrease in PDS avoidance score in patients with permanent ostomy and an increase in closed ostomy, suggesting a possible impact of stoma type (permanent/closed). It remains therefore possible, that in permanent stoma patients, PDS avoidance score could even be reduced after surgery. It can be speculated that there are fewer complications and thus hospitalizations also known to be a triggering factor in PTSD [22]. Furthermore a re-traumatization of patients by anew surgery could be prevented. This especially appeared to be the case in CD patients, indicating a lower susceptibility to PTSD symptoms of permanent stoma patients in general and CD patients in particular. In contrast a in 2019 published US study reported a greater susceptibility of CD patients for PTSD [47]. However no distinction between patients with ostomy and without ostomy and ostomy type (permanent, closed) was made.

Therefore, we may conclude that stoma type may not be the main factor influencing psychological well-being in IBD patients with stoma. However, our data revealed lower IBDQ social function score in stoma vs. non-stoma patients, which is in line with results from other studies indicating lower self-confidence as well as negative perceptions with regards to body image correlating with loneliness in patients after ostomy [48]. Nevertheless in our CD patients this lower score appeared to be improved after stoma application compared to patients not receiving ostomy, indicating a potential beneficial effect on quality of life in this patient group. Overall, our data indicates that specifically CD patients with a solid indication for stoma surgery, may experience a benefit from permanent ostomy.

Moreover, we observed that in stoma patients absenteeism from work was more frequent compared to patients without stoma. This evidently is to be expected, having in mind, that IBD patients in need for stoma formation represent a subgroup of patients with a distinctively severe course of disease. However and most importantly, our results indicate, that this difference in working capacity between patients with vs. without stoma formation was found to be rather minor (with only numerical but not significant overall difference, Fig 2A). Interestingly, a decrease in the fractions of patients actively working with a concomitant increase in patients receiving disability benefits were observed in closed and permanent ostomy patients after surgery. This suggests that the overall negative impact of any stoma formation on capacity to work is substantial. However, only numerically more patients with permanent ostomy were absent from work and the overall difference of working patients in closed vs. permanent stoma patients was rather small. This in conjunction with the overall substantial fraction of almost 20% [24] up to 30% [25] and more of IBD patients overall receiving an invalidity pension on the long-term indicates the following: Although IBD patients in need of a stoma surgery will be at increased risk for permanent work disability and this risk may be higher in patients were a permanent stoma is indicated. However, a substantial fraction of patients even those patients

with permanent stoma formation are capable of continuing their work and the associated negative impact on working capacity in both closed and permanent stoma formation compared to IBD patients without stoma surgery is only moderate. Evidently, further studies are needed to investigate the detailed impact of closed and permanent stoma formation on work disability. This especially holds true in view of the overall limited numbers of patients with stoma surgery in the SIBDCS as well as the potentially remaining selection bias in this study, taking into account, that need for stoma surgery per se may represent one of the strongest indicator (or even surrogate parameter) of a debilitating course of disease. Both these aspects represent limitations of our work. Our work has also several strengths including the nationwide, multicenter prospective inclusion of unselected IBD patients with a long-term follow-up using a multitude of standardized outcome parameters, including a plethora of validated scores regarding psychological wellbeing.

In conclusion, according to our long-term prospective cohort database analysis we observed that IBD patient in need for closed and permanent stoma formation may benefit from this surgical intervention in terms of their luminal and extra-intestinal disease activity. Overall there appears to be only a moderate adverse impact of stoma formation surgery on work disability, even in patients with permanent stoma. Also and in contrary to what one might assume, we did not observe a consistent adverse impact of stoma formation on psychological well-being, neither in closed nor in permanent stoma surgery. Taken together, stoma surgery remains an important tool in the armamentarium of difficult-to-treat IBD patients and our results suggest, that potential downsides in terms of symptoms, quality of life and disability not seldomly feared by patients and potentially their treating physicians alike may be considerably over estimated.

## Supporting information

**S1 Fig. Antibiotics used in UC patients before and after ostomy according to stoma type.**
(TIF)

**S2 Fig. Comparison of mean PDS Avoidance Score in patients with permanent vs. closed ostomy.**
(TIF)

**S3 Fig. PDS Avoidance Score before and after ostomy in CD Patients specifically.**
(TIF)

**S4 Fig. PDS Total Score in CD patients with permanent and closed ostomy before and after ostomy.**
(TIF)

**S5 Fig. IBDQ Social Function Score in CD patients comparing patients with and without ostomy.**
(TIF)

**S6 Fig. IBDQ Social Function Score in patients overall with and without ostomy.**
(TIF)

**S7 Fig. PDS Avoidance Score in patients with vs. without ostomy.**
(TIF)

**S8 Fig. PDS Re-experiencing score in IBD patients with and without ostomy.**
(TIF)

## Acknowledgments

We thank the members of the SIBDCS study group (Claudia Anderegg; Peter Bauerfeind; Christoph Beglinger; Stefan Begré; Dominique Belli; José M. Bengoa; Luc Biedermann; Beat Bigler; Janek Binek; Mirjam Blattmann; Stephan Boehm; Jan Borovicka; Christian P. Braegger; Nora Brunner; Patrick Bühr; Bernard Burnand; Emanuel Burri; Sophie Buyse; Mat-thias Cremer; Dominique H. Criblez; Philippe de Saussure; Lukas Degen; Joakim Delarive; Christopher Doerig; Barbara Dora; Gian Dorta; Mara Egger; Tobias Ehmann; Ali El-Wafa; Mat-thias Engelmann; Jessica Ezri; Christian Felley; Markus Fliegner; Nicolas Fournier; Montser-rat Fraga; Pascal Frei; Remus Frei; Michael Fried; Florian Froehlich; Christian Funk; Raoul Ivano Furlano; Suzanne Gallot-Lavallée; Martin Geyer; Marc Girardin; Delphine Golay; Tanja Grandinetti; Beat Gysi; Horst Haack; Johannes Haarer; Beat Helbling; Peter Hengstler; Den-ise Herzog; Cyrill Hess; Klaas Heyland; Thomas Hinterleitner; Philippe Hiroz; Claudia Hirschi; Petr Hruz; Rika Iwata; Res Jost; Pascal Juillerat; Vera Kessler Brondolo; Christina Knellwolf; Christoph Knoblauch; Henrik Köhler; Rebekka Koller; Claudia Krieger-Grübel; Gerd Kullak-Ublick; Patrizia Künzler; Markus Landolt; Rupprecht Lange; Frank Serge Leh-mann; Andrew Macpherson; Philippe Maerten; Michel H. Maillard; Christine Manser; Michael Manz; Urs Mar-bet; George Marx; Christoph Matter; Valérie McLin; Rémy Meier; Martina Mendanova; Christa Meyenberger; Pierre Michetti; Benjamin Misselwitz; Darius Moradpour; Bernhard Morell; Pat-rick Mosler; Christian Mottet; Christoph Müller; Pascal Müller; Beat Müllhaupt; Claudia Münger-Beyeler; Leilla Musso; Andreas Nagy; Michaela Neagu; Cristina Nichita; Jan Niess; Natacha Noël; Andreas Nydegger; Nicole Obialo; Carl Oneta; Cassandra Oropesa; Ueli Peter; Daniel Peternac; Laetitia Marie Petit; Franziska Piccoli-Gfeller; Julia Beatrice Pilz; Valérie Pittet; Nadia Raschle; Ronald Rentsch; Sophie Restellini; Jean-Pierre Richterich; Sylvia Rihs; Marc Alain Ritz; Jocelyn Roduit; Daniela Rogler; Gerhard Rogler; Jean-Benoît Rossel; Markus Sagmeister; Gaby Saner; Bernhard Sauter; Mikael Sawatzki; Michela Schäppi; Michael Scharl; Martin Schelling; Susanne Schibli; Hugo Schlauri; Sybille Schmid Uebelhart; Jean-François Schnegg; Alain Schoepfer; Frank Seibold; Mariam Seirafi; Gian-Marco Semadeni; David Semela; Arne Senning; Marc Sidler; Christiane Sokollik; Johannes Spalinger; Holger Spangenberger; Philippe Stadler; Michael Steuerwald; Alex Strau-mann; Bigna Straumann-Funk; Michael Sulz; Joël Thorens; Sarah Tiedemann; Radu Tutuian; Stephan Vavricka; Fran-cesco Viani; Jürg Vögtlin; Roland Von Känel; Alain Vonlaufen; Domi-nique Vouillamoz; Rachel Vulliamy; Jürg Wermuth; Helene Werner; Paul Wiesel; Reiner Wiest; Tina Wylie; Jonas Zeitz; Dorothee Zimmermann) for their contribution.

## Author Contributions

**Conceptualization:** Rahel Bianchi, Luc Biedermann.

**Data curation:** Barry Mamadou-Pathé, Jean-Benoit Rossel.

**Formal analysis:** Barry Mamadou-Pathé, Jean-Benoit Rossel.

**Funding acquisition:** Gerhard Rogler.

**Investigation:** Sabine Burk, Babara Dora, Patrizia Kloth.

**Project administration:** Sabine Burk, Babara Dora, Patrizia Kloth.

**Supervision:** Luc Biedermann.

**Writing – original draft:** Rahel Bianchi.

**Writing – review & editing:** Rahel Bianchi, Roland von Känel, René Roth, Philipp Schreiner, Jean-Benoit Rossel, Andreas Rickenbacher, Matthias Turina, Thomas Greuter, Benjamin Misselwitz, Michael Scharl, Gerhard Rogler, Luc Biedermann.

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
