## [Decision Letter · Decision Letter 0]

18 Jul 2022

PONE-D-22-09385Effect of closed and permanent stoma on disease course, psychological well-being and working capacity in Swiss IBD Cohort Study patients.PLOS ONE

Dear Dr. Bianchi,

Thank you for submitting your manuscript to PLOS ONE. After careful consideration, we feel that it has merit but does not fully meet PLOS ONE’s publication criteria as it currently stands. Therefore, we invite you to submit a revised version of the manuscript that addresses the points raised during the review process.

Please submit your revised manuscript by Sep 01 2022 11:59PM. If you will need more time than this to complete your revisions, please reply to this message or contact the journal office at plosone@plos.org. Please include the following items when submitting your revised manuscript:A rebuttal letter that responds to each point raised by the academic editor and reviewer(s). You should upload this letter as a separate file labeled 'Response to Reviewers'.A marked-up copy of your manuscript that highlights changes made to the original version. You should upload this as a separate file labeled 'Revised Manuscript with Track Changes'.An unmarked version of your revised paper without tracked changes. You should upload this as a separate file labeled 'Manuscript'.

We look forward to receiving your revised manuscript.

Kind regards,

Mathilde Body-Malapel

Academic Editor

PLOS ONE

Journal Requirements:

3. One of the noted authors is a group or consortium the Swiss IBD cohort study. In addition to naming the author group, please list the individual authors and affiliations within this group in the acknowledgments section of your manuscript. Please also indicate clearly a lead author for this group along with a contact email address.

Reviewers' comments:

Reviewer's Responses to Questions

**Comments to the Author**

1. Is the manuscript technically sound, and do the data support the conclusions?

Reviewer #1: No

Reviewer #2: Yes

Reviewer #3: Yes

2. Has the statistical analysis been performed appropriately and rigorously? 

Reviewer #1: I Don't Know

Reviewer #2: Yes

Reviewer #3: Yes

3. Have the authors made all data underlying the findings in their manuscript fully available?

Reviewer #1: Yes

Reviewer #2: Yes

Reviewer #3: Yes

4. Is the manuscript presented in an intelligible fashion and written in standard English?

Reviewer #1: No

Reviewer #2: Yes

Reviewer #3: Yes

5. Review Comments to the Author

Reviewer #1: Thank you for inviting me to review this manuscript, but I found this an almost impossible task because of the poor grammar and sentence structure. For example, in the introduction, the sentence starting "The latter representing the standard procedure..." is awkward and difficult to understand.

In the methods section, there is a list of complications but it is unclear what the complications are thought to be of (stoma creation? disease process?) and unclear why the sentence is sitting in the middle of the paragraph.

On line 105, there is a reference to a questionnaire but it isn't clear what questionnaire - although I assume this relates to the IBDQ, the HADS and the PDS which are mentioned later. I am also unclear from the methods when the CDAI and Truelove/Witts were measured.

EIM is mentioned first in the results and not explained.

It appears under the "Course of Disease" heading that CDAI was measured in both Crohns and UC patients.

Finally, I am unclear as to how many of the patients with closed stomas had UC or Crohn's disease and how recurrent disease was measured / taken into account. A panproctocolectomy in someone with UC is, of course, curative, in a way that stoma closure in Crohn's disease may be related to recurrence (in that the anastomosis is usually the site of recurrent disease).

I have not been able to review the results because I do not fully understand the methods as they are currently written.

Having said all that, I'd be very interested/happy to read another draft of the paper.

Reviewer #2: The present study evaluates the long term psychosocial impacts of ostomy in a large Swiss cohort of IBD patients. This is an important area of research with limited existing data, especially over time, and evaluates different types of ostomy IBD patients live with. The study is well designed. Some of the data presentation can be improved upon, especially as it relates to the PDS/psychological well being scores (see below). The discussion needs to be re-written as it appears biased and does not include discussions about the poor outcomes found in this cohort. There is no incorporation of existing data on IBD-related PTSD, which is needed. With these edits the discussion will be more balanced and reflect the prevailing literature as well as the study's actual findings.

I have the following comments/edits that should be considered before publication:

Introduction:

Current estimates of IBD in the US are 3 million versus 1 million. Please amend and update citation.

Please clarify "biological vs. non-biological agents" as medications used to treat IBD.

Sentence starting on line 62 is a bit confusing. Please re-write stating the procedure versus "the latter" for clarity.

Line 78: There are new studies on post-traumatic stress in IBD by Pothemont et al. and Taft et al, please review and cite, especially since discussions of surgery are reported to be potential sources of medical trauma.

Methods:

Line 91: Please update language "suffering from" to "diagnosed with"

Results:

Line 162, line 190: Please update language "suffering from" to "experiencing"

In the table, there is no data for the smoking status line. Is this correct? If so, not sure it's useful to include.

It would be helpful if the data in the "Psychological Well Being" section into a table. As written, it's a bit difficult to follow the numbers pre-post across the different groups.

Line 226: How is the P value for the differences reported 0.99 with a 3-fold increase in PDS score? Please clarify.

Discussion:

The statement that ostomy is not associated with adverse psychological impact conflicts with the findings that those undergoing ostomy report more post-traumatic stress symptoms and poorer social function. PTSD is a significant psychological comorbidity that is chronic without treatment, and likely has larger impacts on patient outcomes than anxiety and depression.

Line 275: Please update language "suffering from" to "diagnosed with"

Section starting on line 304 needs to be re-written. There is robust data regarding PTSD in IBD patients, including how surgery and hospitalizations may be traumatic. These studies need to be incorporated into the present PDS data and discussed.

Reviewer #3: Thank you for the opportunity to review this paper. The authors used a prospectively collected cohort of IBD patients to answer the question of how ostomies affect disease course, psychological well-being, quality of life, and working capacity. They used a propensity score analysis to match patients with and without ostomy. They found lower disease activity in patients after stoma surgery, similar disease-specific QOL, psychological discress, and stress at work in patients with vs without stoma and patients with a permanent stoma vs patients whose stomas were closed. There was a modest adverse effect on working capacity for patients who had a stoma; and a significant reduction in working capacity in Crohn’s patients with a permanent stoma vs stomas that had been closed. My questions are:

1. Did the control group have surgery that did not involve a stoma? Or no surgery at all? Or did some have surgery, some did not, but none of them had a permanent or temporary ostomy?)

2. How did you manage patients with temporary stomas that have not yet been closed in the analyses? Were they considered part of the “permanent stoma” group, or were they not included in the study?

3. On p7, it is stated that patients were eligible if they completed a minimum of one questionnaire before stoma creation, one while they had the stoma, and one after stoma closure. For those who did not have surgery resulting in stoma, what questionnaire completion metrics did they need to meet to be included? Also, are they filling out all of these instruments (IBDQ, HADS, PDS) in one questionnaire, or at least one of the 3 questionnaires?

4. How often was disease severity quantified in this cohort, and did the frequency of disease severity scoring impact eligibility?

5. When you compared control to overall stoma patients, did you compare their scores at that time that they had their stomas (for those who had temporary stomas)? Or after their stomas were closed?

6. For UC patients with stoma compared to those without, there was no difference in disease severity. Presumably these are patients who have had total proctocolectomy with end ileostomy or who had IPAA with temporary stoma compared with patients who have had one-stage IPAA or patients who have not had any surgery at all – please confirm as this is not clear. Were patients with total colectomy with end ileostomy included in this analysis? That is, patients who still had a rectum in place – and thus are halfway through their surgical treatment?

7. For the outcome of medical therapy, are patients with UC who have undergone colectomy or proctocolectomy with stoma being compared to patients who haven’t had surgery and those who had one-stage IPAA? Patients who have had surgery, with or without stoma, for UC would be expected to be off medications completely. When you analyze just Crohn’s disease patients, do you see the same effect?

8. Can the authors comment on why there might be a higher PDS score in patients whose stomas have been closed compared to those with permanent stomas?

6. PLOS authors have the option to publish the peer review history of their article (what does this mean?). If published, this will include your full peer review and any attached files.

Reviewer #1: No

Reviewer #2: No

Reviewer #3: No

---

## [Author Response · Author response to Decision Letter 0]

20 Aug 2022

Manuscript Submission “Effect of closed and permanent stoma on disease course, psychological well-being and working capacity in Swiss IBD Cohort Study patients.”

Dear Editorial Board,

We thank the reviewers and the editorial board very much for their careful consideration of our manuscript. We certainly feel that the suggestions from the reviewer's and the editorial board could significantly improve the manuscript. In addition, we feel, that we could address most of the valuable comments. Please find attached a point-by-point response with referral to the respective changes in the manuscript which we attached in track change form. Again, we would like to take this opportunity to thank the journal and the reviewers for considering our work.

We are happy to respond to any potential further questions.

In the name of all authors, Rahel Bianchi and Luc Biedermann

 

Reviewer: 1

Comments to the Author

Thank you for inviting me to review this manuscript, but I found this an almost impossible task because of the poor grammar and sentence structure. For example, in the introduction, the sentence starting "The latter representing the standard procedure..." is awkward and difficult to understand.

We thank the reviewer for this comment. According to his suggestion on English language we performed a careful review of the entire manuscript and performed several adaptations in terms of language and wording which are apparent in track change mode of the manuscript. As the sentence. "The latter representing the standard procedure..." was not only difficult to read for this reviewer but for reviewer two as well we agree that it is very difficult to read and have therefore applied changes specifying the procedure. Otherwise, and despite the concerns of Reviewer#1 which we of course kindly acknowledge, we would kindly like to add, that amongst the writers, there are numerous experienced clinical and basic scientists with respective experience in English and scientific writing. Moreover, this “almost impossible” task neither appeared to be equally difficult for Reviewer#2 and #3 nor the co-authors. 

In the methods section, there is a list of complications but it is unclear what the complications are thought to be of (stoma creation? disease process?) and unclear why the sentence is sitting in the middle of the paragraph.

We thank the reviewer for this important and well-taken comment. We fully agree, that we missed the occasion to make it entirely clear to the reader, what the exact nature of these described complications was. In this case, we were referring to a composite of a group of major complications associated with CD and UC per se. We have changed the manuscript accordingly (line 183 and following paragraph) and changed the text to reduce confusion about the definition of complications in the middle of the paragraph. 

On line 105, there is a reference to a questionnaire but it isn't clear what questionnaire - although I assume this relates to the IBDQ, the HADS and the PDS which are mentioned later. 

To gather data for the Swiss IBD Cohort Study patients and physicians are asked to fill out a number of questionnaires including medical (diagnosis, smoking status, pregnancy, current severity of disease, clinical course, past and current therapy, adverse events of therapy, supplementation therapy, laboratory analysis, flare ups and triggering factor, new exams and outcomes, disease complications, disease location, surgery, extraintestinal manifestations, fissures, abscesses, fistula and stenosis,), psychological (IBDQ, HADS, PDS, Efford-Reward-Ratio) and work related data (working status, absence of work, disability pension) once a year. Indeed, these questionnaires include the validated scores, mentioned by Reviewer#1. Having said that, we agree with the reviewer that he term “Questionnaire” is not clear. Therefore, we have specified it in line 191. 

I am also unclear from the methods when the CDAI and Truelove/Witts were measured.

We agree with the reviewer that this is not only a crucial point but not sufficiently addressed in the methods section. CDAI and MTWAI were measured when the patients were included in the study as well as in the annual follow ups. We have added this information to the manuscript at line 209 and 210 and thank the reviewer for pointing out this issue. 

EIM is mentioned first in the results and not explained.

We thank the reviewer again for this important comment. We completely agree that EIM have to be mentioned in the methods and explained. We changed the manuscript accordingly and appropriately introduced the term (line 180).

It appears under the "Course of Disease" heading that CDAI was measured in both Crohns and UC patients.

This is indeed an important remark as the text in the manuscript does not sufficiently distinguish between the two groups and therefore one could come to the assumption that CDAI was used for CD and UC patients. We have changed the manuscript accordingly at line 275 and 279. 

Finally, I am unclear as to how many of the patients with closed stomas had UC or Crohn's disease and how recurrent disease was measured / taken into account. A panproctocolectomy in someone with UC is, of course, curative, in a way that stoma closure in Crohn's disease may be related to recurrence (in that the anastomosis is usually the site of recurrent disease).

We thank the reviewer for this important comment and provided information about all the groups including the patients with closed stoma in figure one. In patients with closed stoma 47.7% had CD and 52.3% were diagnosed with UC which is likewise mentioned in the manuscript at line 309. Therefore, this important piece of information as mentioned by Reviewer#1 is readily available. Recurrent disease was not taken into account in the CD group when performing the analysis as we wanted to provide a sufficient cohort size and there was no scheduled, mandatory or standardized procedure to investigate for recurrence of disease. This represents one limitation of a cohort study design, as compared to a prospective protocol-guided investigation. However, we feel that the reviewer mentions a crucial fact in this comment regarding a high recurrence rate of Crohn’s disease especially in ileostomy which should be discussed as it provides one potential explanation, as to why no difference could be observed when comparing disease activity in UC patients with permanent or closed stoma (a respective paragraph has been included in the discussion starting at line 433). 

Reviewer: 2

Comments to the Author

Current estimates of IBD in the US are 3 million versus 1 million. Please amend and update citation.

We thank the reviewer for this important comment. We have updated the introduction accordingly using a more current and less conservative estimate of the US patient numbers including a respective reference (line 124). 

Please clarify "biological vs. non-biological agents" as medications used to treat IBD.

We would like to thank the reviewer for pointing out this relevant difficulty to understand both terms and their exact meaning. We have therefore specified the terms with examples on line 129 and 130.

Sentence starting on line 62 is a bit confusing. Please re-write stating the procedure versus "the latter" for clarity.

We agree with Reviewer #1 and #2 who both suggested to change this sentence as it is difficult to read. As we would like to provide an easy readable text to readership we made the appropriate changes to the manuscript text (now line 134). 

Line 78: There are new studies on post-traumatic stress in IBD by Pothemont et al. and Taft et al, please review and cite, especially since discussions of surgery are reported to be potential sources of medical trauma.

We again thank the reviewer for this comment regarding post traumatic stress in IBD patients and encouraging us to include important work on post-traumatic stress to be incorporated in this paper. Indeed, these studies undermine the importance of surgery and medical procedures in general for the development of post-traumatic stress symptoms. Therefore, we have included these studies in the introduction (Pothemont et al., Patient Perspectives on Medical Trauma Related to Inflammatory Bowel Disease, J Clin Psychol Med Settings, 2021) (lines 153-157) as well as the discussion (Pothemont et al., Patient Perspectives on Medical Trauma Related to Inflammatory Bowel Disease, J Clin Psychol Med Settings, 2021, Taft et al. Posttraumatic Stress in Patients with Inflammatory Bowel Disease: Prevalence and relationships to Patient-Reported Outcomes, Inflamm Bowel Dis, 2022 and Taft et al. Initial Assessment of Post-traumatic Stress in an US Cohot of Inflammatory Bowel Disease Patients, Inflamm Bowel Dis, 2019) starting at line 478. 

Line 91: Please update language "suffering from" to "diagnosed with"

Line 162, line 190: Please update language "suffering from" to "experiencing"

Line 275: Please update language "suffering from" to "diagnosed with"

These suggested changes have been incorporated in the manuscript.

In the table, there is no data for the smoking status line. Is this correct? If so, not sure it's useful to include.

The reviewer is correct that there is no data in the evaluated group in our table. This is due to the fact, that the smoking status during surgery procedures, was not optimally assessed. We therefore decided to entirely delete this from our table and thus updated table one accordingly. 

It would be helpful if the data in the "Psychological Well Being" section into a table. As written, it's a bit difficult to follow the numbers pre-post across the different groups.

We would like to thank the reviewer for this comment. We are fully aware that it can be difficult especially as mentioned when switching between the different groups. We therefore provided several figures especially for psychological wellbeing and work ability as we felt this was an easy to comprehend way to show our data. We also added a paragraph summarizing the results starting at line 371. However, if a table is preferred we are more than happy to provide all the data accordingly. 

Line 226: How is the P value for the differences reported 0.99 with a 3-fold increase in PDS score? Please clarify.

We thank the reviewer for this comment as we agree that it is difficult to understand what is shown in the text. The roughly 3-fold increase refers to the PDF PRIOR ostomy and here, for this value we found a significant difference. However, after ostomy, the PDS between the two groups is roughly similar. Although the median value appears robustly different. The confidence interval is the IQR is similar and thus also no significant differences (p-value 0.99) was found. 

The statement that ostomy is not associated with adverse psychological impact conflicts with the findings that those undergoing ostomy report more post-traumatic stress symptoms and poorer social function. PTSD is a significant psychological comorbidity that is chronic without treatment, and likely has larger impacts on patient outcomes than anxiety and depression.

Section starting on line 304 needs to be re-written. There is robust data regarding PTSD in IBD patients, including how surgery and hospitalizations may be traumatic. These studies need to be incorporated into the present PDS data and discussed.

We thank the reviewer for this important annotation. We fully agree with the reviewer that it is crucial to incorporate the newest data provided especially by Taft et al. in the discussion as it shows clearly in multiple publications the negative impact of surgery, hospitalization and severity of the disease on PTSD symptoms. We have therefore included several studies starting at line 478 (Pothemont et al., Patient Perspectives on Medical Trauma Related to Inflammatory Bowel Disease, J Clin Psychol Med Settings, 2021, Taft et al. Posttraumatic Stress in Patients with Inflammatory Bowel Disease: Prevalence and relationships to Patient-Reported Outcomes, Inflamm Bowel Dis, 2022 and Taft et al. Initial Assessment of Post-traumatic Stress in an US Cohot of Inflammatory Bowel Disease Patients, Inflamm Bowel Dis, 2019) and discussed them in them in the manuscript. We also would like to add that we were aware that surgery can be a traumatic event, however our data does not suggest stoma type (permanent/closed) to be a factor in PTSD formation.

Reviewer: 3

Comments to the Author

Did the control group have surgery that did not involve a stoma? Or no surgery at all? Or did some have surgery, some did not, but none of them had a permanent or temporary ostomy?)

We thank Reviewer#3 for this important question. With regards to UC patients it is easy to answer, as in Switzerland no colonic resection in UC patient (virtually all will be total proctocolectomy; only rarely colectomy with rectal sparing) is performed without a temporary (sometimes even a permanent) stoma application; either as a (modified) two-step or more frequently three step procedure. With regards to CD patients undergoing segmental intestinal resection the situation is a bit more complex indeed. Virtually all patients with emergent resection in view of a penetrating complication (fistula formation and/or abscess) will receive a temporary stoma. In contrast, segmental resection – most often performed for fibrostenosis or refractory inflammatory and mixed fibro-inflammatory stenosis – in a more elective fashion is performed in the majority of cases without stoma. We feel, that these patients would not represent an ideal control group, as patients with for instance an ileal fibrostenosis undergoing ileocecal resection typically represent another and rather distinctively different group of patients. 

How did you manage patients with temporary stomas that have not yet been closed in the analyses? Were they considered part of the “permanent stoma” group, or were they not included in the study?

Again, this is an important point. Indeed, it lies in the nature of the definition of temporary vs. permanent stoma formation, that any temporary ostomy might one day be reverted. For instance, this is often a remaining hope for patients with refractory high burden peri-anal CD having undergone diversion ostomy. Could one call this a permanent stoma e.g. 2 years after the initial surgical step or is there a hope, that with emerging medical therapy options, one day, the stoma might be reverted and thus in hindsight would be characterized as a temporary stoma?

Evidently, the latter is possible. In our study, every patient with sustained ostomy present at follow-up after initial stoma formation surgery was considered as “permanent” stoma in our analysis. 

On p7, it is stated that patients were eligible if they completed a minimum of one questionnaire before stoma creation, one while they had the stoma, and one after stoma closure. For those who did not have surgery resulting in stoma, what questionnaire completion metrics did they need to meet to be included? Also, are they filling out all of these instruments (IBDQ, HADS, PDS) in one questionnaire, or at least one of the 3 questionnaires?

How often was disease severity quantified in this cohort, and did the frequency of disease severity scoring impact eligibility?

For those who did not have surgery resulting in stoma, what questionnaire completion metrics did they need to meet to be included? 

We thank the reviewer for these important questions and are happy to clarify them further. The IBDQ, HADS and PDS are all included in one questionnaire and have to be fully completed by the patient without their treating physician (PRO). Disease severity was assessed at the minimum for a least three times for all patients. Disease severity is also measured by a questionnaire every time psychological scores are evaluated as well. The follow up questionnaires were sent to patients and their physicians once a year and had to be completed at least at year 1, 3 and 5 for eligibility in this study. The frequency of disease severity scoring therefore did not impact eligibility. We agree that the inclusion criteria especially for the control group is not explained thoroughly enough. We added the information therefore at lines 205-207. 

When you compared control to overall stoma patients, did you compare their scores at that time that they had their stomas (for those who had temporary stomas)? Or after their stomas were closed?

We thank the reviewer again for this comment as we are happy to clarify, that we compared the scores at the time the stoma was in place. We agree that this is not fully clear, we adapted the manuscript accordingly. (Line 206)

For UC patients with stoma compared to those without, there was no difference in disease severity. Presumably these are patients who have had total proctocolectomy with end ileostomy or who had IPAA with temporary stoma compared with patients who have had one-stage IPAA or patients who have not had any surgery at all – please confirm as this is not clear. Were patients with total colectomy with end ileostomy included in this analysis? That is, patients who still had a rectum in place – and thus are halfway through their surgical treatment?

We thank the Reviewer#3 again for this important remark. As mentioned above in our answer to a previous comment: there is no one-stage IPAA performed in Switzerland. Indeed, patients undergoing a presumptive 3 step proctocolectomy or colectomy with rectum left in place (as mentioned above, this only rarely performed in Switzerland, if at all, virtually always in young females with child-bearing potential) and without further continuing the procedure of pre-designated stoma formation but preference of sustained ileostomy (with or without rectum in place) were included here. In our experience, there is a small fraction of patient deciding to refrain from continuation of surgical step half-way through.

For the outcome of medical therapy, are patients with UC who have undergone colectomy or proctocolectomy with stoma being compared to patients who haven’t had surgery and those who had one-stage IPAA? 

As mentioned above, there is no comparison to one-stat IPAA. We however do agree with Reviewer#3 that this is a relevant question. Our cohort study however, is not ideal to study such potential differences in outcome and patient preferences between one, two and three step IPAA procedure. In the last years, there first has been a trend suggesting a potential advantage of three-step procedure, while more recently two step procedures - above all the modified version – may have a comparable outcome and low fraction of complications at least in tertiary referral centers. The latter might be one of the main concerns for extrapolation of favourable modified two-step data. Again, the SIBDCS certainly is not the ideal format to investigate this highly important and relevant issue. 

Patients who have had surgery, with or without stoma, for UC would be expected to be off medications completely. When you analyze just Crohn’s disease patients, do you see the same effect?

We thank the reviewer for this comment. The comparison between CD patients with permanent and closed stoma we can observe a decrease of medication use in both group without a significant difference between them. 

Can the authors comment on why there might be a higher PDS score in patients whose stomas have been closed compared to those with permanent stomas?

We would like to thank the reviewer for this comment but unfortunately it was not entirely clear to us to which specific PDS score Reviewer#3 was primarily aiming to in the question. As we suppose it was PDS avoidance score we attribute the higher score in patients with closed stoma to possible post-operative complication and re-experiencing the trauma from the first surgery when the stoma was formed again. We agree with the reviewer that this has not been addressed sufficiently. The manuscript has been changed accordingly (starting at line 495). 

Yours sincerely,

Rahel Bianchi and Luc Biedermann in the name of all coauthors

---

## [Editor Report · Decision Letter 1]

2 Sep 2022

Effect of closed and permanent stoma on disease course, psychological well-being and working capacity in Swiss IBD Cohort Study patients.

PONE-D-22-09385R1

Dear Dr. Bianchi,

We’re pleased to inform you that your manuscript has been judged scientifically suitable for publication and will be formally accepted for publication once it meets all outstanding technical requirements.

Kind regards,

Mathilde Body-Malapel

Academic Editor

PLOS ONE
---

## [Editor Report · Acceptance letter]

7 Sep 2022

PONE-D-22-09385R1 

Effect of closed and permanent stoma on disease course, psychological well-being and working capacity in Swiss IBD Cohort Study patients. 

Dear Dr. Bianchi:

I'm pleased to inform you that your manuscript has been deemed suitable for publication in PLOS ONE. Congratulations! Your manuscript is now with our production department. 

Kind regards, 

on behalf of

Dr. Mathilde Body-Malapel 

Academic Editor

PLOS ONE